# Prognostic Significance of PD-L1 Expression on Circulating Myeloid-Derived Suppressor Cells in NSCLC Patients Treated with Anti-PD-1/PD-L1 Checkpoint Inhibitors

**DOI:** 10.3390/ijms252212269

**Published:** 2024-11-15

**Authors:** Roser Salvia, Laura G. Rico, Teresa Morán, Jolene A. Bradford, Michael D. Ward, Ana Drozdowskyj, Joan Climent-Martí, Eva M. Martínez-Cáceres, Rafael Rosell, Jordi Petriz

**Affiliations:** 1Functional Cytomics Lab, Germans Trias i Pujol Research Institute (IGTP), Campus Can Ruti, Crta. de Can Ruti, Camí de les Escoles, s/n, 08916 Badalona, Spain; rsalvia@igtp.cat (R.S.); lauragarciarico91@gmail.com (L.G.R.); 2Department of Cellular Biology, Physiology and Immunology, Autonomous University of Barcelona (UAB), 08193 Cerdanyola del Vallès, Spain; jcliment.germanstrias@gencat.cat (J.C.-M.); emmartinez.germanstrias@gencat.cat (E.M.M.-C.); 3Medical Oncology Department, Catalan Institute of Oncology Badalona (ICO), Germans Trias i Pujol University Hospital (HUGTiP), 08916 Badalona, Spain; mmoran@iconcologia.net; 4Applied Research Group in Oncology, Germans Trias i Pujol Research Institute (IGTP), 08916 Badalona, Spain; 5Department of Medicine, Autonomous University of Barcelona (UAB), 08193 Cerdanyola del Vallès, Spain; 6Thermo Fisher Scientific, Fort Collins, CO 80524, USA; jolene.bradford@thermofisher.com (J.A.B.);; 7Translational Cancer Research Unit, Dr. Rosell Oncologic Institute, Dexeus University Hospital, 08028 Barcelona, Spain; anadroz@gmail.com (A.D.); rrosell@oncorosell.com (R.R.); 8Immunology Division, Germans Trias i Pujol University Hospital, 08916 Badalona, Spain; 9 Germans Trias i Pujol Research Institute (IGTP), 08916 Badalona, Spain

**Keywords:** PD-L1, immunotherapy, NSCLC, flow cytometry

## Abstract

Even though anti-PD-1/PD-L1 immune checkpoint inhibitors (ICIs) in non-small cell lung cancer (NSCLC) have improved survival, a high percentage of patients still do not respond to ICIs. Myeloid-derived suppressor cells (MDSCs) are circulating cells that express PD-L1 and can infiltrate and proliferate in the tumor microenvironment, inducing immunosuppression. By evaluating changes in PD-L1 expression of live peripheral blood MDSCs, we are able to define a new PD-L1 index, useful in predicting ICI escape in NSCLC patients before initiating anti-PD-1/PD-L1 immunotherapy. In this study, a cohort of 37 NSCLC patients was prospectively analyzed, obtaining independent PD-L1 indexes. In patients with a PD-L1 index > 5.88, progressive disease occurred in 58.33% of patients [median progression-free survival (PFS) = 5.73 months; 95%CI = 2.67–20.53], showing significant differences when compared with patients with a PD-L1 index ≤ 5.88, in whom 7.69% progressed and median PFS was not reached (NR); *p*-value = 0.0042. Overall survival (OS) was significantly worse in patients with a high vs. low PD-L1 index (41.67% vs. 76.92%; median OS = 18.03 months, 95%CI = 6.77–25.23 vs. NR, 95%CI = 1.87-NR; *p*-value = 0.035). The PD-L1 index can be applied to stratify NSCLC patients according to their probability of response to ICIs at baseline. In addition to quantifying tumoral expression, this index could be used to compare nonresponse to treatment.

## 1. Introduction

Lung cancer is the leading cause of cancer death in the entire world population [1,2]. In the last decade, the approval of immune checkpoint inhibitors (ICIs), such as anti-PD-1/PD-L1, has improved survival and long-term responses in patients with non-small cell lung cancer (NSCLC) without anaplastic lymphoma kinase (ALK) or epidermal growth factor receptor (EGFR) driver mutations [3,4,5,6]. Although considerable progress has been made in reducing lung cancer mortality, only around 20% of lung cancer patients benefit long term from ICIs, according to data from randomized clinical trials comparing ICIs in monotherapy and combined with chemotherapy to the standard chemotherapy in different clinical scenarios [3,4,5,6]. Currently, the only biomarker approved to guide the prescription of ICIs in different NSCLC clinical scenarios is the tumor-proportional score (TPS) of PD-L1 in the tumor, despite its limited predictive value [7,8,9,10]. Given the potential adverse effects of immunotherapy and the cost of these therapies, new predictive biomarkers beyond PD-L1 expression in the tumor are needed to better select candidates most likely to benefit from such therapies.

Myeloid-derived suppressor cells (MDSCs) are circulating immature cells that express PD-L1 and induce immunosuppression [11]. Under physiological conditions, immature myeloid cells lack immune-suppressive activity and reside mainly in the bone marrow. Signals derived from trauma, infections or cancer impair the monocyte and neutrophil developmental pathway, favoring the switch to an immune-suppressive immature state, the MDSCs. These cells express increased levels of reactive oxygen, nitric oxide, arginase-1, PGE2 and cytokines [11,12,13]. Evidence suggests that MDSC are upregulated in cancer [14], such as in NSCLC, where MDSCs are pathologically expanded and associated with resistance to ICIs [15,16,17].

This study describes the use of a new biomarker index, the PD-L1 index, which predicts the response to ICIs of NSCLC patients at baseline. This index is calculated by considering the differential reactivity against the PD-L1 protein in MDSCs and is measured by flow cytometry in live cells, representing a rapid and minimally invasive approach that provides information on the clinical outcome of patients before initiating anti-PD-1/PD-L1 treatment.

## 2. Results

### 2.1. Patients Characteristics

From March 2020 to May 2022, 37 patients with NSCLC who attended the Medical Oncology Department at the Catalan Institute of Oncology (ICO) Badalona (Spain) were included in this study after signing their informed consent to participate. The date of the final follow-up session was 4 November 2022. Before initiating treatment with ICIs, an EDTA-anticoagulated peripheral blood sample was obtained and analyzed to determine the circulating MDSC population. Baseline demographic data and clinical and molecular data are detailed in Table 1.

For the ICI response control study, only 11 patients were analyzed due to death or loss to follow-up. An additional group of NSCLC patients (*n* = 12) is included in the follow-up study, resulting in a total group of 23 patients.

### 2.2. Clinical Outcome

At the end of this study, ten patients had died due to progression of disease: two from heart failure; two from respiratory infection; one from pneumonia, one from myositis, myocarditis and myasthenia gravis overlap syndrome (adverse events related to the ICI); and one due to fatal hemoptysis (tumor-related disease), as presented in Table 2. The median duration of follow-up was 13.3 months (range, 5 days to 32.2 months).

Overall, the median duration of ICIs was 4.6 months (range, 1 day to 24.8 months). Five patients (13.5%; four in the first-line ICI group with pembrolizumab and one in the consolidation ICI group with durvalumab) were censored, as they were still receiving their assigned treatment at the end of this study.

ORR for the entire cohort was 56.7% (all patients presented PR). SD was observed in 10.8% with a disease control rate of 67.5%.

Nine out of thirty-seven patients (24.3%) completed the proposed ICI (2 years of first-line ICI in five patients (13.5%) and 1 year of ICI after chemoradiation (CTRT) in four (10.8%)) and have not progressed. Two additional patients who exhibited immune-related adverse events (general toxicity and renal failure related to ICI plus CT) that required treatment discontinuation present a long-lasting response in the follow-up.

The median PFS and OS for the entire cohort were 20.5 (95% confidence interval [95%CI] = 5.7-Not Reached [NR]) and 25 (95%CI = 12.8-NR) months, respectively. The median PFS and OS were 20.5 (95%CI = 5-NR) months and NR (95%CI = 6.8-NR) in patients treated with first-line ICI; 15.3 (95%CI = 5.7-NR) and 25.2 (95%CI = 18-NR) for patients who received first-line chemotherapy (CT) plus an ICI; 2.2 (95%CI = 1.5-NR) and 4.6 (95%CI = 1.4-NR) for patients treated in second-line ICI, and both NR (95%CI = NR-NR) for patients who received an ICI as consolidation after CTRT, respectively.

According to tissue PD-L1 expression, PFS and OS were 2.7 (95%CI = 2.1-NR) and 9.9 (95%CI = 4.6-NR) months for 0% PD-L1 expression, both NR (NR-NR) for 1–49%, and 20.5 (95%CI = 5.3-NR) months and NR (6.8-NR) for ≥50%, respectively.

### 2.3. PD-L1 Index of Myeloid-Derived Suppressor Cells Is a Predictor of Clinical Outcome

The PD-L1 index of MDSCs defined in this study takes into account the mean fluorescence intensity (MFI) of stimulated cells normalized by the MFI of unstimulated cells (see the formula in Section 4.2 and Appendix A), and assesses the reactivity of the PD-L1 molecule after ex vivo stimulation. The predictive value of the PD-L1 index in MDSCs at baseline was assessed by ROC curve analysis (Figure 1). Youden’s index allowed the identification of an optimal cut-off point (with a sensitivity of 93.3% and a specificity of 54.5%), as well as to classify the present cohort of NSCLCs into two identifiable groups based on their PD-L1 index values (low PD-L1 index group ≤ 5.88; and high PD-L1 index group > 5.88).

The low PD-L1 index group had an ORR of 76.9%, while in the high PD-L1 index group the ORR was 45.8%. Differences in PFS and OS of the two groups were estimated using the Kaplan–Meier method (Figure 2 and Appendix A). For those patients with a PD-L1 index > 5.88, PD occurred in 14 out of 24 patients (58.33%; PFS median = 5.73 months; 95%CI = 2.67–20.53), showing significant differences when comparing with patients with a PD-L1 index ≤ 5.88, where 1 of 13 patients (7.69%) progressed and median PFS was NR (95%CI = NR-NR) with a *p*-value = 0.0042 (Figure 2A). OS was significantly worse in patients with a high (41.67%; median OS of 18.03 months, 95%CI = 6.77–25.23) vs. low PD-L1 index (76.92%; median OS NR, 95%CI = 1.87-NR), with a *p*-value = 0.035 (Figure 2B). The low PD-L1 index group significantly benefited from anti-PD-1/PD-L1 treatment in terms of higher PFS (Figure 2A) and OS (Figure 2B) compared to the high PD-L1 index group.

Univariate and multivariate Cox proportional hazards regression models were determined to identify factors influencing PFS and OS (Appendix A), sex, age, smoking habit, number of metastases, cancer staging, histology, ECOG performance status, LDH levels, driver mutations, tumor PD-L1, and PD-L1 index. In the univariate Cox model, groups based on the PD-L1 index were associated with a significant impact on PFS and OS (*p*-value = 0.0212 and 0.0462, respectively), as were duration of anti-PD-1/PD-L1 therapy and sex (Figure 3). There is a sex imbalance in the patients included (73% male), which reflects the clinical practice in our center. In our setting, more men than women continue to be diagnosed and treated for lung cancer. In Figure 3A, males have a hazard ratio of 13.44 over females (*p*-value = 0.0143), indicating that there is a higher risk of disease progression in males, although the aforementioned gender imbalance should be considered. Remarkably, PD-L1 expression in tumor tissue was not found relevant to predict the PFS nor OS (Figure 3A and Appendix A). However, when stratified in 0% vs. ≥1%, the 0% PD-L1-expressing tumors had a significantly better PFS and OS (*p*-value = 0.021 and 0.028; Appendix A). Furthermore, the stratification in 0%, 1–49% and ≥50% showed the inconsistency of this tissue biomarker, as 1–49% PD-L1^+^ tumors presented a better outcome than ≥50% PD-L1^+^ tumors (Appendix A). No association was found between the PD-L1 expression in tissue and the PD-L1 index, and the combination of both variables could not better define the outcome to anti-PD-1/PD-L1 therapy (Appendix A). Additionally, the combination of the PD-L1 index with the presence of driver mutations did not predict the outcome (Appendix A). In the multivariate model, only duration of therapy was relevant for OS.

Follow-up of the NSCLC patients at a median of 4.3 months showed that the PD-L1 index increased significantly from 7.71 ± 2.90 at baseline to 9.72 ± 3.75 at follow-up (*p*-value = 0.0135; Figure 4). Interestingly, when classifying patients according to clinical outcome, there was a difference between progressive and nonprogressive patients. In patients who progressed (*n* = 12), there was a significant increase in the PD-L1 index, from 8.51 ± 2.78 at baseline to 11.50 ± 4.15 at follow-up (*p*-value = 0.0122), whereas in patients who did not progress, there were no significant differences in the PD-L1 index after ICI treatment (baseline, 6.84 ± 2.78; follow-up, 7.77 ± 1.83; *p*-value = 0.3652). This result links the increase in the PD-L1 index in response control with a poor prognosis.

## 3. Discussion

PD-L1 protein expression in the tumor, assessed by immunohistochemistry, is the currently designated biomarker for therapeutic decision-making in NSCLC. Typically, PD-L1 expression above 50% translates into better anti-PD-1/PD-L1 treatment performance [18,19,20]. However, the literature shows controversial results on PD-L1 expression and clinical outcome [21,22,23,24,25], which may hinder the predictive value of tumor PD-L1 testing. The expression of PD-L1 in tumor tissues can vary widely [26], and the ability to measure PD-L1 levels is dependent on the availability of tumor tissue. In some patients, the amount of tumor tissue may be limited, which can compromise the ability to measure PD-L1 levels. Therefore, the use of complementary methods consisting of the use of alternative minimally invasive tests [27] may be of particular interest in these patient groups for refining therapeutic decisions.

In fact, some biomarkers have been linked to the success of ICIs. Tumor mutational burden has been associated with the efficacy of nivolumab and ipilimumab in NSCLC [28]. In addition, an effector T-cell gene signature in the tumor defined as the expression of CXCL9, PD-L1 and IFN-γ messenger RNA has been proposed as possible predictive marker of ICI efficacy [29]. Other biomarkers defined in NSCLC are the neutrophil-to-lymphocyte ratio (NLR) [30,31], derived NLR (dNLR) [32], lung immune prognostic index (LIPI), which is based on dNLR and LDH [33,34], or lung immuno-oncology prognostic score-3 (LIPS-3) [35], although none of these stands as a consistent biomarker in our cohort (Appendix A). Appendix A provides information about the different outcomes based on treatment strategy. 

Our study defines a new promising biomarker: the PD-L1 index. Assessed in live, circulating MDSCs by flow cytometry, it provides information on PFS and OS in stage III-IV NSCLC. In our cohort of patients, progressive disease occurred in 58.3% of patients in the high PD-L1 index group, while 7.7% progressed in the low PD-L1 index group (*p*-value = 0.0042). In terms of OS, 76.9% of low-index patients survived, while only 41.7% of high-index patients survived (*p*-value = 0.035). The PD-L1 index is a valuable predictor of PFS and OS by univariate Cox regression, along with ICI duration and sex. As previously described, there is cumulative evidence of sex differences in predictors of response to ICI, although the clinical outcome remains less clear [36,37,38,39]. Specifically, males tend to have better responses to ICI monotherapy, whereas females benefit from chemotherapy plus ICI treatment [36], but in this case, the sex imbalance in the cohort hinders this correlation.

In addition, the levels of the PD-L1 index are consistent with the control of response in terms of clinical outcome. In line with previous results, an increase in the PD-L1 index from follow-up compared to the basal PD-L1 index is associated with a poor prognosis (*p*-value = 0.0122), indicating that patients with progressive disease tend to have a higher unfolding of PD-L1 in MDSCs after anti-PD-1/PD-L1 treatment.

As we have shown, reactivity to PD-L1 in circulating MDSCs increases dramatically and differentially after ex vivo stimulation of these cells. The observed dynamic changes in PD-L1 are very relevant, especially because PD-L1 seems to be hidden in circulating MDSCs and its expression can only be revealed by ex vivo stimulation with PMA. This phenomenon could be associated with conformational changes after stimulation.

PD-L1 has a complex glycosylation and related function that may have an important impact on immunotherapy resistance [40]. PD-L1 homodimerization and mutations cannot be ruled out [41], so the biological and clinical significance of these reactivity changes will require further investigation. It would be crucial to determine why PD-L1 can always remain hidden, i.e., not expressed after stimulation with PMA, and if confirmed, whether these patients could have poorer clinical outcomes to anti-PD-1/PD-L1 treatment. Importantly, only patients who benefit from anti-PD-1/PD-L1 antibody therapy have a PD-L1 index below the described cutoff point after stimulation.

In this study, PD-L1 expression in tumor tissue did not correlate with the PD-L1 index identified in MDSCs. This suggests that adaptive mechanisms of PD-L1 expression may be at play in these circulating cells. However, it is also possible that the intratumoral heterogeneity reflects the limitations of routine clinical tumor biopsy. Further studies are needed to clarify the underlying mechanisms of PD-L1 expression in MDSCs.

This noninvasive peripheral blood analysis coupled with a rapid flow cytometry protocol the same day after drawing could help to refine decision-making on ICIs in the clinical setting, which currently relies on the PD-L1 protein expression in tumor tissue in advanced NSCLC patients. This is because the PD-L1 index, which can be quantified from peripheral blood, could be used to predict treatment response in addition to the PD-L1 protein expression in tumor tissue.

Additionally, future studies are needed to extend the application of the PD-L1 index to other solid tumors as well as to other clinical scenarios in lung cancer where the ICIs are being approved, such as adjuvant and neoadjuvant settings in NSCLC and advanced SCLC.

Limitations of this study are related to the quantification of the number of circulating MDSCs at baseline and during treatment and, as might be expected, the population of MDSCs in blood would decrease in parallel with treatment response and duration of response. The type of MDSCs, PMN-MDSCs and M-MDSCs could perhaps have some relevance.

## 4. Materials and Methods

### 4.1. Patients

This prospective cohort study included 37 adult patients diagnosed with stage III and IV NSCLC and previously eligible to receive anti-PD-1/PD-L1 treatment (see Table 1 for a full description of clinical indications). An additional cohort of 12 patients was included in the follow-up study. PD-L1 expression was analyzed using the Ventana PD-L1 Assay^®^ (SP263) in the tumor sample. Molecular characterization of the tumor was performed by NGS Oncomine Comprehensive Assay, 161 genes (Life Technologies^®^, Carlsbad, CA, USA).

Inclusion criteria included indication for anti-PD-1/PD-L1 treatment, availability for clinical follow-up and willingness to undergo a blood draw for research.

Clinical visits were scheduled according to clinical practice, and computed tomography scans to assess radiographic response during treatment were performed every 6–8 weeks, according to clinical practice. All patients were followed until the end of this study, death, withdrawal of consent or loss of follow-up. Two patients who died before response assessment were excluded from the analysis.

### 4.2. Sample Preparation and Flow Cytometry Data Analysis

Detailed information on the flow cytometry procedure for analyzing PD-L1 reactivity in MDSCs is presented here [42]. This minimally invasive method represents a rapid assessment of PD-L1, which seeks to preserve the native structure of the molecule using living cells the same day after drawing. Reactivity against PD-L1 was analyzed by quantifying the mean fluorescence intensity of PD-L1 within the MDSC population when comparing unstimulated and stimulated specimens.

First, peripheral blood samples were collected in EDTA-anticoagulated tubes and prepared with minimal sample perturbation protocols within 4 h of venous extraction [43,44]. Briefly, 100 µL of unlysed whole blood were diluted in 900 µL Hanks’ balanced salt solution supplemented with 1% bovine serum albumin and 0.1% sodium azide in duplicate. Both tubes were incubated with Vybrant™ DyeCycle™ Violet (DCV; 50 µM final concentration; Invitrogen^TM^, Eugene, OR, USA) and 10 µL fetal bovine serum at 37 °C for 10 min in a dedicated water bath, protected from light.

After incubation, blood was stimulated with 1 µL phorbol 12-myristate 13-acetate (PMA; 1.63 µM final concentration) in one tube and 1 µL of DMSO as a negative control in other tube. Blood specimens were stimulated for 5 min at 37 °C in a dedicated water bath protected from light.

After stimulation, the tubes were spun down from 0 to 16,100 rcf in 10 s and 900 µL of supernatant were reserved. The remaining 100 µL were used for immunophenotyping by adding 5 µL of PE-PD-L1, PE-Cy7-CD33, FITC-HLA-DR, and APC-CD11b (Invitrogen™, Eugene, OR, USA) at a final concentration of 2.5 µg/mL, and were incubated for 20 min at room temperature and protected from light. Then, labeled samples were diluted with the reserved supernatant and 2 µL of 7-AAD (stock solution 10 µg/mL; 15.74 nM final concentration) were added. After 5 min of incubation, samples were immediately acquired on the Attune NxT™ Flow Cytometer (Thermo Fisher Scientific, Eugene, OR, USA), at a flow rate of 25 µL/min to ensure an event rate not exceeding 400 events/s. To ensure the significance of the results, a minimum of 20,000 nucleated cells (defined by nucleated DCV^+^ cells) were acquired. Threshold levels were set empirically using violet SSC (V-SSC) versus DCV dual plot to eliminate debris and the large number of erythrocytes.

FCS files obtained on the flow cytometer were analyzed with FlowJo™ (FlowJo LLC, Ashland, OR, USA). First, nonnucleated events (DCV^neg^), necrotic cells (7-AAD^+^), doublets and aggregates were discarded from the analysis. Next, MDSCs were selected as HLA-DR^low/−^CD33^+^CD11b^+^ cells and PD-L1 mean fluorescence intensity (MFI) was assessed in stimulated samples (PMA) and unstimulated samples (DMSO). An example of this gating strategy is shown in Appendix A. The PD-L1 index was calculated using the MFI of PD-L1 in the MDSCs population as follows:$$PDL1index=\frac{{MFI}_{PMA}-{MFI}_{DMSO}}{2\times {SD}_{DMSO}}$$
where MFI is the mean fluorescence intensity; SD is the standard deviation; PMA is the PD-L1^+^ MDSCs incubated in the presence of PMA; and DMSO is the PD-L1^−^ MDSCs incubated in the presence of DMSO.

### 4.3. Statistical Analysis

Quantitative variables were described as means, medians, SDs and ranges. Qualitative variables were described as absolute frequencies and percentages. Efficacy data were evaluated in terms of overall response rate (ORR), progression-free survival (PFS) and overall survival (OS). Overall response rate (ORR) was calculated according to Response Evaluation Criteria in Solid Tumors (RECIST) 1.1 as complete response (CR), partial response (PR), stable disease (SD), progressive disease (PD) or nonassessable (NA).

The receiver operating characteristic (ROC) curve was calculated to measure the accuracy of the PD-L1 index in predicting PFS. The optimal cutoff point was obtained from the Youden index [45] and allowed us to classify patients into two groups. From these groups, PFS and OS survival curves were generated using the Kaplan–Meier method [46]. PFS was defined as the time from diagnosis to progression, death or last follow-up date. OS was defined as the time from diagnosis to death from any cause or last follow-up. Patients who were still alive at the date of the last follow-up were censored. Differences between survival curves were tested by the log-rank test. Cox proportional hazards regression models were used to evaluate patient- and tumor-related factors. Variables that showed a *p*-value ≤ 0.1 in the univariate linear regression analysis were included in the multivariate analysis. The Kaplan–Meier method was performed in RStudio (v. 2021.09.2) and SAS (v. 9.4) and the Cox proportional hazards regression analysis and forest plots in SAS (v. 9.4). Differences between follow-up and baseline PD-L1 index were calculated using the Wilcoxon matched-pairs signed rank test with GraphPad Prism (v. 9.0). Values were considered significant when *p* ≤ 0.05.

## 5. Conclusions

Our discovery suggests that MDSCs play a key role in the efficacy of ICI treatments. This is because the number of MDSCs in the blood can be used to predict how well a patient will respond to treatment. Importantly, new therapies could be developed that specifically target MDSCs. This could be carried out by controlling the pathological expansion of MDSCs or by unhiding PD-L1 antigen in MDSCs. In conclusion, the PD-L1 index is a rapid and powerful tool to predict disease progression at baseline and to anticipate the efficacy of anti-PD-1/PD-L1 treatment.

## Figures and Tables

**Figure 1 ijms-25-12269-f001:**
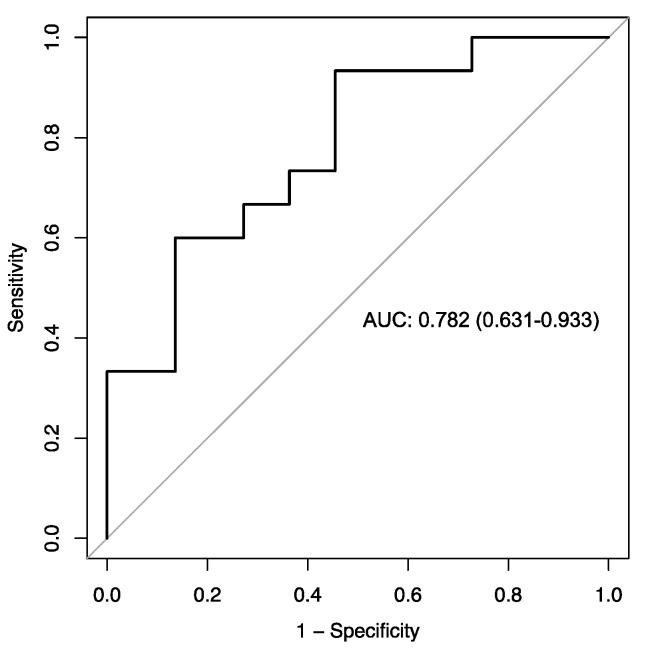
Receiver operating characteristic (ROC) curve of PD-L1 index in MDSCs. According to ROC curve analysis, the cutoff value of 5.88 was confirmed as the PD-L1 index value to predict progression-free survival in the cohort of NSCLC patients (with 93.3% sensitivity and 54.5% specificity). ROC curve analysis (area under the curve, AUC = 0.782; 95%CI = 0.63–0.93) classified two identifiable groups of patients at baseline according to MDSC expression of a PD-L1 index ≤ 5.88 (low PD-L1 index group, *n* = 13 patients) and >5.88 (high PD-L1 index group, *n* = 24 patients).

**Figure 2 ijms-25-12269-f002:**
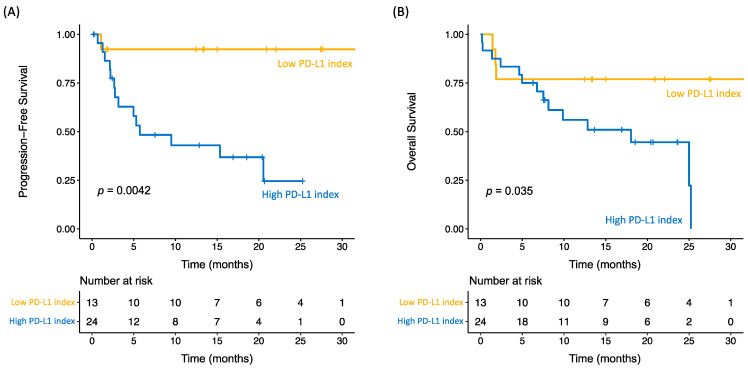
Kaplan–Meier curves of progression-free survival and overall survival at baseline. (**A**) For those patients in the low PD-L1 index group, PD occurred in 1 out of 13 patients (7.69%) and the median progression-free survival (PFS) was not reached (NR), 95%CI = NR-NR, showing significant differences when comparing to patients in the high PD-L1 index group, where 14 out of 24 patients experienced PD (58.33%; median PFS = 5.73 months, 95%CI = 3.17-NR), with a *p*-value = 0.0042. (**B**) Overall survival (OS) was significantly worse in patients with a high (41.67%; median OS of 18.03 months, 95%CI = 6.77–25.23) vs. low PD-L1 index (76.92%; median OS NR, 95%CI = 1.87-NR), with a *p*-value = 0.035.

**Figure 3 ijms-25-12269-f003:**
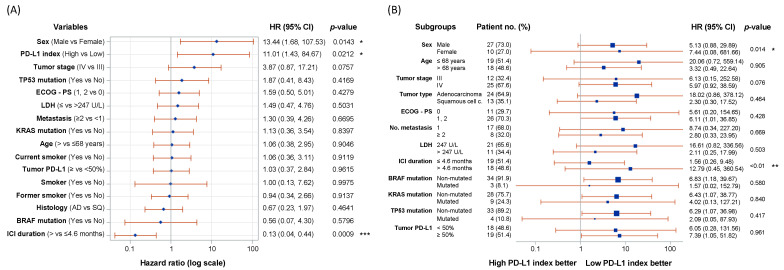
Impact of patient’s characteristics on progression-free survival. (**A**) Forest plot displaying the impact of patient’s characteristics, duration of ICI and PD-L1 index groups on PFS. Considering that higher hazard ratio (HR) values imply more risk to disease progression, male patients and patients within the high PD-L1 index group presented worse PFS, whereas patients receiving an ICI for more than 4.6 months had better PFS. HR had 95% Wald Confidence Limits and were calculated with Cox regression. (**B**) Forest plot showing the risk of PD-L1 index groups to undergo disease progression, displayed for each key variable subgroup. In the case of sex, age, tumor type, LDH, ICI duration, and BRAF mutation, one of the subgroups had a better clinical outcome and few disease-progression events. Therefore, this lack of events led to poor statistical power and resulted in high confidence intervals. This subgroup analysis was calculated using the Firth correction. *p*-value was considered significant when <0.05 (*), <0.01 (**) and <0.001 (***), or alternatively non-significant. Legend: ECOG-PS, Eastern Cooperative Oncology Group performance-status score; LDH, lactate dehydrogenase; AD, adenocarcinoma; SQ, squamous cell carcinoma.

**Figure 4 ijms-25-12269-f004:**
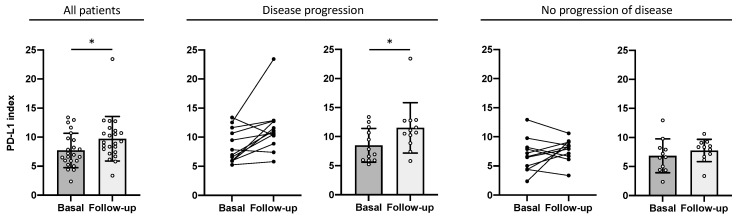
The PD-L1 index from baseline to response follow-up. The PD-L1 index was analyzed before the administration of immunotherapy (basal) and after a median of approximately 4 months (follow-up). The first graph shows that in all patients (*n* = 23), the PD-L1 index levels at follow-up were higher than at baseline (*p*-value = 0.0135). Those patients who progressed (*n* = 12) had a higher PD-L1 index at follow-up than at baseline (*p*-value = 0.0122), while those who did not progress (*n* = 11) had similar levels of the PD-L1 index (*p*-value = 0.3652). Statistics were calculated using the Wilcoxon matched-pairs signed rank test, where *p*-value was considered significant if <0.05 (*).

**Table 1 ijms-25-12269-t001:** Demographic, clinical and molecular characteristics of patients at baseline.

Characteristic	Value
Patients enrolled—no. (%)	37 (100%)
Male sex—no. (%)	27 (73%)
Median age—year (range)	68 (50–83)
Histology—no. (%)	
Adenocarcinoma	24 (64.9%)
Squamous cell carcinoma	13 (35.1%)
Driver or actionable mutations—no. (%)	
TP53	3 (8.1%)
KRAS	9 (24.3%)
BRAF	3 (8.1%)
None	22 (59.5%)
Tumor PD-L1 expression—no. (%)	
0–20%	18 (48.7%)
21–49%	0 (0%)
50–60%	4 (10.8%)
61–70%	6 (16.2%)
71–80%	7 (18.9%)
81–90%	2 (5.4%)
Stage at diagnosis—no. (%)	
IIIA	3 (8.1%)
IIIB	9 (24.3%)
IV	25 (67.6%)
ECOG performance-status score—no. (%)	
0	11 (29.7%)
1	24 (64.9%)
2	2 (5.4%)
Therapeutic Strategies ^(a)^—ICI type, no. (%)	
First-line ICI	Pembrolizumab, 15 (40.5%)
First-line chemotherapy plus ICI	Pembrolizumab, 6 (16.2%)
Second-line ICI	Pembrolizumab, 2 (5.4%) Nivolumab, 6 (16.2%) Atezolizumab, 1 (2.7%)
Consolidation ICI after chemoradiation	Durvalumab, 7 (18.9%)
History of tobacco use—no. (%)	
Never	2 (5.4%)
Former	24 (64.9%)
Current	11 (29.7%)

ECOG = Eastern Cooperative Oncology Group. ^(a)^ ICI treatment details: Patients included in the present cohort received ICIs according to the following criteria: (1) first-line pembrolizumab monotherapy (2 mg/kg/21 days or 4 mg/kg/42 days up to 24 months) for patients with squamous and non-squamous histology and PD-L1 expression ≥50% in tumor specimen; (2) first-line platinum-doublet chemotherapy with cisplatin (75 mg/m^2^/21 days up to four cycles) or carboplatin (AUC5/21 days up to four cycles) and pemetrexed (500 mg/m^2^/21 days up to progression) plus pembrolizumab (2 mg/kg/21 days up to 24 months) for patients with non-squamous histology and PD-L1 expression ≤ 49%; (3) second-line ICI with nivolumab (3 mg/kg/15 days or 6 mg/kg/28 days up to progression), pembrolizumab (2 mg/kg/21 days or 4 mg/kg/42 days up to progression) or atezolizumab (1200 mg/21 days up to progression); and (4) consolidation immunotherapy with durvalumab (10 mg/kg/15 days up to 12 months) in stage III patients after chemoradiation and PD-L1 expression ≥ 1%.

**Table 2 ijms-25-12269-t002:** Characteristics of patients’ clinical outcomes one year after initiation of anti-PD-1/PD-L1.

Characteristic	Value
Best response (RECIST criteria)—no. (%)	
Partial response	21 (56.7%)
Stable disease	4 (10.8%)
Progressive disease	8 (21.6%)
Nonassessable	4 (10.8%)
Median ICI duration—months (range)	
First-line ICI	7.5 (0–24.8)
First-line chemotherapy plus ICI	9.5 (0.8–21)
Second-line ICI	0.8 (0–2.8)
Consolidation ICI after chemoradiation	11 (1.4–12.7)
Disease progression—no. (%)	
No progression	22 (59.5%)
Progression	15 (40.5%)
Survival status—no. (%)	
Alive	20 (54%)
Deceased	17 (46%)
Death causes—no. (%)	
Disease progression	10 (27%)
Respiratory disease ^(a)^	3 (8.1%)
Cardiac failure	2 (5.4%)
ICI-related adverse events ^(b)^	1 (2.7%)
Tumor-related disease ^(c)^	1 (2.7%)

RECIST, Response Evaluation Criteria in Solid Tumors; ICI, immune checkpoint inhibitor. ^(a)^ Respiratory tract infection (2 patients, 5.4%) and pneumonia (1 patient, 2.7%). ^(b)^ Myositis, myocarditis and myasthenia gravis overlap syndrome. ^(c)^ Fatal hemoptysis.

## Data Availability

Data sets generated during the current study are available from the corresponding author on reasonable request.

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
