# Peer review of "Prognostic Significance of PD-L1 Expression on Circulating Myeloid-Derived Suppressor Cells in NSCLC Patients Treated with Anti-PD-1/PD-L1 Checkpoint Inhibitors"

_ijms, 2024, doi:10.3390/ijms252212269_

Round 1

Reviewer 1 Report

Comments and Suggestions for Authors

The manuscript by Salvia et al, describes a possible method to improve treatment of NSLC patients. The methodology is well described, although figures can be improved. Supplementary data is missing and the ppt file I received has figures but no legends, and in the last figure (C), histogram lines are missing. Precisely this figure should be included in the article to better understand the methodology, because readers can be not get used to flow cytometry. Some abbreviatures needs description by the same reason but in these case to clinic nomenclature.

Author Response

Dear Editor in Chief,

We sincerely appreciate your thoughtful and insightful valuable suggestions on our original submission. We carefully considered all of your remarks, which, along with the reviewers' valuable suggestions, have resulted in a significant improvement in the manuscript. Please find below our comprehensive responses to their comments.

Sincerely,

Jordi Petriz.

REVIEWER 1

The manuscript by Salvia et al, describes a possible method to improve treatment of NSCLC patients.

  • The methodology is well described, although figures can be improved.
  • Supplementary data is missing and the ppt file I received has figures but no legends, and in the last figure (C), histogram lines are missing. Precisely this figure should be included in the article to better understand the methodology, because readers can be not get used to flow cytometry.
  • Some abbreviatures needs description by the same reason but in these case to clinic nomenclature.

We appreciate the valuable comments on the manuscript and apologize for not including the supplementary data, which were inadvertently omitted when uploading the various documents during the submission process. The figures have now been edited as follows: The color palette in Figure 2 has been changed to avoid the green/red colors that were confusing. Figure 3 has also been edited to improve the labels and resolution, and its footnote has been changed accordingly. Supplementary Figure S2 has been added and referenced in the manuscript (line 212) to better understand the methodology and concept of the PD-L1 index. Supplementary Figure S3 has been added to line 227. The meaning of the abbreviations CTRT and CT (chemoradiation and chemotherapy) was missing and has been changed.

Reviewer 2 Report

Comments and Suggestions for Authors

Dear authors,

After reviewing this study, I have several concerns:

1. The subject population of both gender quite skewed. Gender disparities of clinical response to ICI is well-documented (PMID 37760403). In Figure 3A, gender was suggested as a critical variance to affect PFS. Therefore, supplying female subjects to balance the gender ratio, or presenting data by gender seems to be mendatory.

2. The rationale of using circulating PD-L1-expressed MDSC as a biomarker for predicting ICI response is not sufficient. Applying PD-1/PD-L1 immunotherapy includes TPS (https://www.merck.com/product/usa/pi_circulars/k/keytruda/keytruda_pi.pdf) whereas which can not properly predict the clinical response to immunotherapies (PMID 36781219). Such information can help readers understanding 

3. Some sentence can be more precise. For example, "Although considerable progress has been made in reducing lung cancer mortality, only a small proportion of patients benefit from ICI" at Line 50 - 52. What is the precise number of "a small proportion"?

Author Response

Dear Editor in Chief,

We sincerely appreciate your thoughtful and insightful valuable suggestions on our original submission. We carefully considered all of your remarks, which, along with the reviewers' valuable suggestions, have resulted in a significant improvement in the manuscript. Please find below our comprehensive responses to their comments.

Sincerely,

Jordi Petriz.

REVIEWER 2:

  1. The subject population of both gender quite skewed. Gender disparities of clinical response to ICI is well-documented (PMID 37760403). In Figure 3A, gender was suggested as a critical variance to affect PFS. Therefore, supplying female subjects to balance the gender ratio, or presenting data by gender seems to be mandatory.

We fully agree that there is cumulative evidence of sex differences in clinical response to ICI. The current cohort includes consecutive patients identified at our center, representing the population we see in our outpatient clinic (more males than females).

Response to Q1:  We appreciate the reviewer’s suggestion, and a detailed sex-based dataset has now been included in the Results section, which reads as follows:

Results, line 252: There is a sex imbalance in the patients included (73% male), which reflects the clinical practice in our center. In our setting, more men than women continue to be diagnosed and treated for lung cancer. In Figure 3A, males have a hazard ratio of 13.44 over females (p-value = 0.0143), indicating that there is a higher risk of disease progression in males, although the aforementioned gender imbalance should be considered.

Discussion, line 385: The PD-L1 index is a valuable predictor of PFS and OS by univariate Cox regression, along with ICI duration and sex. As previously described, there is cumulative evidence of sex differences in predictors of response to ICI, although the clinical outcome remains less clear. Specifically, males tend to have better responses to ICI monotherapy, whereas females benefit from chemotherapy plus ICI treatment [1–3] but in this case the sex imbalance in the cohort hinders this correlation.

  1. The rationale of using circulating PD-L1-expressed MDSC as a biomarker for predicting ICI response is not sufficient. Applying PD-1/PD-L1 immunotherapy includes TPS (https://www.merck.com/product/usa/pi_circulars/k/keytruda/keytruda_pi.pdf) whereas which cannot properly predict the clinical response to immunotherapies (PMID 36781219). Such information can help readers understanding.

We fully agree that PD-L1 TPS is an imperfect biomarker for selecting ICI in the clinical setting. However, despite the research effort to find novel biomarkers to redefine the clinical indication of ICI, PD-L1 TPS are currently the only approved biomarkers that guide ICI prescription in lung cancer patients.

Response to Q2: We have clarified this point by rephrasing the sentence as follows:

Currently, the only biomarker approved to guide the prescription of ICI in different NSCLC clinical scenarios is the tumor-proportional score (TPS) of PD-L1 in the tumor, despite its limited predictive value (Chen et al., 2024; Davis & Patel, 2019; Evans et al., 2018; Nikolic et al., 2023). Given the potential adverse effects of immunotherapy and the cost of these therapies, new predictive biomarkers beyond PD-L1 expression in the tumor are needed to better select candidates most likely to benefit from such therapies.

Related to this, we have expanded the results section by adding the supplementary figures S4-S14 to better demonstrate the need of an additional biomarker (line 257 and line 314):

Remarkably, PD-L1 expression in tumor tissue was not found relevant in predicting PFS or OS (Figure 3A, Figure S4). However, when stratified at 0% vs ≥1%, tumors with 0% PD-L1 expression had significantly better PFS and OS (p-value=0.021 and 0.028; Figure S5). Furthermore, stratification into 0%, 1-49% and ≥50% showed the inconsistency of this tissue biomarker, as 1-49% PD-L1+ tumors presented better outcome than ≥50% PD-L1+ tumors (Figure S6). No association was found between tissue PD-L1 expression and PD-L1 index, and the combination of both variables could not better define outcome to anti-PD-1/PD-L1 therapy (Figure S7-S8). Additionally, the combination of PD-L1 index with the presence of driver mutations did not predict outcome (Figure S9).

Other biomarkers defined in NSCLC are the neutrophils-to-lymphocyte ratio (NLR)[4,5], derived NLR (dNLR)[6], lung immune prognostic index (LIPI)[7,8], or the lung immuno-oncology prognostic score-3 (LIPS-3)[9], although none of these stands as a consistent biomarker in our cohort (Figure S10-S13).

  1. Some sentence can be more precise. For example, "Although considerable progress has been made in reducing lung cancer mortality, only a small proportion of patients benefit from ICI" at Line 50 - 52. What is the precise number of "a small proportion"?

According to data from RCT in different clinical scenarios and different ICI in lung cancer patients (see as example REF 3 to 6), around 20% of the patients receiving such treatment benefit on the long term.

Response to Q3:  This information has been included in the Introduction (line 52) and now reads as follow:

Although considerable progress has been made in reducing lung cancer mortality, only about 20% of the lung cancer patients benefit long-term from ICI, according to data form randomized clinical trials comparing ICI in monotherapy and combined with chemotherapy to the standard chemotherapy in different clinical scenarios [3-6].

Added bibliography:

  1. Lang, D.; Brauner, A.; Huemer, F.; Rinnerthaler, G.; Horner, A.; Wass, R.; Brehm, E.; Kaiser, B.; Greil, R.; Lamprecht, B. Sex-Based Clinical Outcome in Advanced NSCLC Patients Undergoing PD-1/PD-L1 Inhibitor Therapy-A Retrospective Bi-Centric Cohort Study. Cancers (Basel) 2021, 14, doi:10.3390/CANCERS14010093.
  2. Suay, G.; Garcia-Cañaveras, J.-C.; Aparisi, F.; Lahoz, A.; Juan-Vidal, O. Sex Differences in the Efficacy of Immune Checkpoint Inhibitors in Neoadjuvant Therapy of Non-Small Cell Lung Cancer: A Meta-Analysis. Cancers (Basel) 2023, 15, 4433–4447, doi:10.3390/cancers15184433.
  3. Liang, J.; Hong, J.; Tang, X.; Qiu, X.; Zhu, K.; Zhou, L.; Guo, D. Sex Difference in Response to Non-Small Cell Lung Cancer Immunotherapy: An Updated Meta-Analysis. Ann Med 2022, 54, 2605–2615, doi:10.1080/07853890.2022.2124449.
  4. Nindra, U.; Shahnam, A.; Stevens, S.; Pal, A.; Nagrial, A.; Lee, J.; Yip, P.Y.; Adam, T.; Boyer, M.; Kao, S.; et al. Elevated Neutrophil-to-Lymphocyte Ratio (NLR) Is Associated with Poorer Progression-Free Survival in Unresectable Stage III NSCLC Treated with Consolidation Durvalumab. Thorac Cancer 2022, 13, 3058–3062, doi:10.1111/1759-7714.14646.
  5. Mandaliya, H.; Jones, M.; Oldmeadow, C.; Nordman, I.I.C. Prognostic Biomarkers in Stage IV Non-Small Cell Lung Cancer (NSCLC): Neutrophil to Lymphocyte Ratio (NLR), Lymphocyte to Monocyte Ratio (LMR), Platelet to Lymphocyte Ratio (PLR) and Advanced Lung Cancer Inflammation Index (ALI). Transl Lung Cancer Res 2019, 8, 886–894, doi:10.21037/TLCR.2019.11.16.
  6. Mezquita, L.; Preeshagul, I.; Auclin, E.; Saravia, D.; Hendriks, L.; Rizvi, H.; Park, W.; Nadal, E.; Martin-Romano, P.; Ruffinelli, J.C.; et al. Predicting Immunotherapy Outcomes under Therapy in Patients with Advanced NSCLC Using DNLR and Its Early Dynamics. Eur J Cancer 2021, 151, 211–220, doi:10.1016/J.EJCA.2021.03.011.
  7. Aldea, M.; Benitez, J.C.; Mezquita, L. The Lung Immune Prognostic Index (LIPI) Stratifies Prognostic Groups in Advanced Non-Small Cell Lung Cancer (NSCLC) Patients. Transl Lung Cancer Res 2020, 9, 967–970, doi:10.21037/TLCR.2020.04.14.
  8. Mezquita, L.; Auclin, E.; Ferrara, R.; Charrier, M.; Remon, J.; Planchard, D.; Ponce, S.; Ares, L.P.; Leroy, L.; Audigier-Valette, C.; et al. Association of the Lung Immune Prognostic Index With Immune Checkpoint Inhibitor Outcomes in Patients With Advanced Non–Small Cell Lung Cancer. JAMA Oncol 2018, 4, 351–357, doi:10.1001/JAMAONCOL.2017.4771.
  9. Banna, G.L.; Cortellini, A.; Cortinovis, D.L.; Tiseo, M.; Aerts, J.G.J.V.; Barbieri, F.; Giusti, R.; Bria, E.; Grossi, F.; Pizzutilo, P.; et al. The Lung Immuno-Oncology Prognostic Score (LIPS-3): A Prognostic Classification of Patients Receiving First-Line Pembrolizumab for PD-L1 ≥ 50% Advanced Non-Small-Cell Lung Cancer. ESMO Open 2021, 6, 100078, doi:10.1016/J.ESMOOP.2021.100078.

Minor comments:

The affiliation from “Universitat Autònoma de Barcelona” in line 7 has been changed to its English version.

We have added “— no. (%)” in the Histology box in Table 1 and unified the nomenclature of first-line ICI, second-line ICI in tables 1 and 2 with a hyphen.

In line 236, text has been added to clarify the description of PD-L1 index (“the MFI of unstimulated cells”).

In line 273, the p-value of the PD-L1 index for PFS and OS has been added (p-value = 0.0212 and 0.0462 respectively).

In Figure 4, the color has been changed to a grey palette and the word analyzed in the footnote has been changed to analyzed to match American English.

In the bibliography, 3 references were missing their doi number and this has been added. We have noticed that the reference 14 was missing from the list (Salvia, R.; Rico, L.G.; Ward, M.D.; Bradford, J.A.; Petriz, J. Functional Flow Cytometry to Predict PD-L1 Conformational Changes. Curr Protoc 2023, 3, e944, doi:10.1002/cpz1.944.). We have added this reference and it is now assigned to number 18.

Round 2

Reviewer 1 Report

Comments and Suggestions for Authors

Manuscript is acceptable for publishing

Author Response

We would like to thank the Reviewer for taking the time and effort necessary to review the manuscript. We sincerely appreciate all valuable comments and suggestions, which helped us to improve the quality of the manuscript.

Reviewer 2 Report

Comments and Suggestions for Authors

Dear authors,

Thank you for your kindly response to my concerns. Those response relieve my concerns. Only one minor suggestion:

In Line 325 - 327 of revised manuscript, the author described ". As previously described, there is cumulative evidence of sex differences in predictors of response to ICI, although the clinical outcome remains less clear." The citation of "As previously described" is missing. Regarding "although the clinical outcome remains less clear", the authors can refer to two references (PMID 39317781 and 37993681) which can provide insights to the underlying mechanism of gender disparities in ICI response.

Except for the above minor suggestion, I have no further concerns.

Author Response

We would like to thank the Reviewer 2 for taking the time and effort necessary to review the manuscript. We sincerely appreciate all valuable comments and suggestions, which helped us to improve the quality of the manuscript.

We have moved the citations to the first sentence and we have the added one of the references you proposed to better demonstrate these sex differences on clinical outcomes.    "As previously described, there is cumulative evidence of sex differences in predictors of response to ICI, although the clinical outcome remains less clear [42–45]. Specifically, males tend to have better responses to ICI monotherapy, whereas females benefit from chemotherapy plus ICI treatment [44], but in this case the sex imbalance in the cohort hinders this correlation."